



# Mist Cannon Trucks Can Exacerbate Secondary Organic Aerosol Formation and PM$_{2.5}$ Pollution in the Road Environment

Yu Xu[1], Xin-Ni Dong[2], Chen He[3], Dai-She Wu[4], Hong-Wei Xiao[1], Hua-Yun Xiao[1,*]

[1]School of Environmental Science and Engineering, Shanghai Jiao Tong University, Shanghai 200240, China

[2]Jiangxi Province Science and Technology Information Institute, Nanchang 330000, China

[3]State Key Laboratory of Heavy Oil Processing, China University of Petroleum, Beijing 102249, China

[4]School of Resource, Environmental and Chemical Engineering, Nanchang University, Nanchang 330031, China

*Corresponding author: Hua-Yun Xiao

E-mail: xiaohuayun@sjtu.edu.cn

Phone: +86-173-0183-7060





**Abstract:** Mist cannon trucks have been widely applied in megacities in China to
reduce the road dust. Their practical effect on controlling the formation of secondary
organic aerosol and fine particles remains unknown. We characterized the chemical
composition variations in $PM_{2.5}$ collected on the road sides with the simulated
operations of mist cannon truck and traditional sprinkling truck via Fourier transform
ion cyclotron resonance mass spectrometry and ion chromatography. The mass
concentrations of water-soluble organic carbon in $PM_{2.5}$ showed a significant increase
(62-70%) after air spraying. Further, we found that secondary organic aerosols,
particularly organic nitrates, increased significantly via the interactions of reactive gas-
phase organics, atmospheric oxidants, and aerosol liquid water after air spraying,
although the air spraying had a better effect on suppressing road dust than the ground
aspersion. Moreover, the formation of $PM_{2.5}$ in the road segment where the mist cannon
truck passed was promoted, with an increase of up to 13% in mass concentration after
25–35 minutes, on average. The application of mist cannon trucks undoubtedly worsens
the road atmospheric environment and causes health hazards to walking residents. The
overall results provide not only valuable insights to the formation processes of
secondary organic aerosols associated with aerosol liquid water in the road environment
but also management strategies to regulate the mist cannon truck operation in China.

**Keywords:** Mist cannon truck; Water mist; Secondary organic aerosols; $PM_{2.5}$; Process
and mechanism



## 1 Introduction


Over the past decade, the demand for effective road dust control has grown
dramatically due to the upgraded environmental protection policies and quality of life.
Traditionally, the sprinkling trucks with the ground aspersion work well for vehicle-
generated road dust. The newly developed mist cannon trucks are able to spray water
mist up to 120 meters away and 100 m high, with a droplet diameter of as small as 5
μm. They are considered to be more water-saving and efficient than the traditional
sprinkling truck (the ground aspersion). Thus, the mist cannon trucks have been widely
utilized by the local environmental bureau in recent years to achieve the target of strict
emission control in megacities in China (Wang et al., 2022). Traffic-related emissions
contribute a huge amount of volatile organic compounds (VOCs), nitrogen oxides
($NO_x$), ammonia ($NH_3$), and fine aerosol particles ($PM_{2.5}$) to the urban atmosphere,
which exerts adverse impacts on human health and climate change (Deng et al., 2020;
Yang et al., 2022). However, no study has investigated whether and how the water mist
sprayed by mist cannon truck affects the road atmospheric environment in coordination
with traffic emissions.
As we know, the mist cannon trucks can spray a large amount of fine water mist in
a short time. It is expected that the air humidity of local road environment where the
mist cannon truck passed will increase sharply. Aerosol liquid water (ALW) exists in
the condensed phase as a function of particle chemical composition, particle
concentration, temperature (T), and relative humidity (RH) (Nguyen et al., 2015; Xu et
al., 2020a). Typically, an increase in RH can promote the rise in ALW concentration





(Guo et al., 2015). ALW, as a ubiquitous and abundant medium, can not only facilitate
partitioning of gas-phase water-soluble organics to the condensed phase but also drive
the formation of secondary organic aerosol (SOA) (Carlton and Turpin, 2013; Sareen
et al., 2017). The severe haze episodes in Beijing can even be partly attributed to the
interactions between ALW (or high RH) and aerosol organic components (Li et al., 2019;
Wang et al., 2021). In particular, 40–80% of fossil-fuel-derived primary organic
aerosols were found to be water-soluble (Qiu et al., 2019). Undoubtedly, there are large
knowledge gaps in our current understanding on ALW-related water-soluble SOA
formation in the road environment with mist cannon truck operation.

Although few studies have systematically evaluated the ability of mist cannon

truck to remove road dust, it is easy to understand that the tiny water droplets generated
by mist cannon system can indeed capture coarse particles (i.e., dropping dust to the
ground) more effectively than the water column sprayed by the traditional sprinkling
truck. However, fine particles (e.g., $PM_{2.5}$) are a major threat to urban atmospheric
environment and human health (Yue et al., 2020). Thus, it is necessary to understand
the removal effect of mist cannon truck on fine particles in road environment. Assessing
the impact of the mist cannon truck operation on road $PM_{2.5}$ pollution is also of great
significance for guiding future environmental protection initiatives.

In this study, we simulated the operation scenes of mist cannon truck and

traditional sprinkling truck on the sides of the urban road (Nanchang, eastern China)
and collected ambient $PM_{2.5}$ samples in these scenes. The molecular compositions of
water-soluble organic matter (WSOM) in $PM_{2.5}$ samples were resolved using ultrahigh-



resolution Fourier transform ion cyclotron resonance mass spectrometry (FT-ICR MS).
We also present the measurements of the relevant chemical parameters in $PM_{2.5}$ samples
and the predicted ALW concentration. Furthermore, the variations of $PM_{2.5}$
concentrations in the road segment where the mist cannon truck passed were monitored.
The obtained results will clarify the impact of spraying water mist by mist cannon truck
on SOA formation and $PM_{2.5}$ pollution control in the urban road environment for the
first time.

**2 Experimental Section**
**2.1 Study Site and Sample Collection.**

A branch road (provincial capital north 2nd road) that was selected as study area

was located in the centre of Nanchang (Eastern China) (**Figure S1a**). This area is
characterized by heavy traffic and high population density. There are no typical
pollution sources, such as factories and garbage treatment plants, within 30 km of the
study area. The trees on both sides of the road are very high and luxuriant (**Figure S1a**),
likely indicating that the atmosphere in this road environment is rarely disturbed by
strong winds and is relatively stable. The dominant species of the trees at the area is
camphor trees (*Cinnamomum Camphora*). Thus, the region is expected to be influenced
by both anthropogenic (vehicle exhausts) and biogenic VOCs.

Two sampling points with a distance of approximately 70 m were selected, which

were respectively on the roadside effected by air spray and ground aspersion (**Figure**
**S1b-e**). The air spraying 8 m above the ground was to simulate the water mist sprayed
by the mist cannon truck. In contrast, the ground aspersion with a height of 0.4 m above


the ground was designed to simulate the operation of traditional sprinkling truck.
Typically, thousands of oral fluid droplets emitted by loud speech will disappear in the
time range of 8 to 14 minutes in the stagnant air environment (Stadnytskyi et al., 2020).
It implied that the residence time of the fine water mist sprayed by the mist cannon
truck in the air may be longer than 8 min. As mentioned above, the luxuriant trees on
both sides of the road cause the atmosphere in the road environment to be relatively
stable with less disturbance by strong winds. Thus, the frequency of spraying water was
set to 1 minute spraying every 8 minutes (lower limit) in this study. The spraying was
controlled by an intelligent timing irrigation equipment (Nadster, China). It should be
noted that the diameter of the pores in the nozzle for air spraying is less than 0.8 mm,
while that in the nozzle for ground aspersion is approximately 6 mm.

Fine aerosol particles ($PM_{2.5}$) were collected onto the prebaked (500°C for ~10

hours) quartz fiber filters (Pallflex, Pall Corporation, USA) using a high-volume air
sampler (Series 2031, Laoying, China). Sampling at the above-mentioned two sites was
simultaneously performed from 23 March to 26 March, 2021. The duration of each
aerosol sampling was approximately 4 h (9:00−13:00 LT) every day. We observed that
the traffic flow on March 25 was higher than that on other days. The weather of
sampling periods was cloudy to sunny, sunny, sunny, and shower (only one short
precipitation event in the sampling period), corresponding an ambient average
temperature (T) of approximately 21 °C. The ambient average relative humidity (RH)
in those periods (23–25 March) ranged between 50% and 60%. The average RH can
reach 84–87% after air spraying. The ambient average RH was as high as 80% during





the period of 9:00−13:00 on 26 March. Thus, the water spraying operation was stopped
when the samples were collected on 26 March. The samples were stored at −28 °C prior
to the analysis. In addition, the mass concentrations of $PM_{2.5}$ were measured online
(Thermo Scientific 5030i, USA) from July to August 2021 near the trunk road where
the mist cannon truck frequently operated (**Figure S1a**). Specifically, $PM_{2.5}$
concentrations near road (81 road, Nanchang) were recorded at 5-minute intervals from
10 minutes before to 50 minutes after the mist cannon truck passed by.

**2.2 Chemical Analysis.**

A portion of each filter sample was ultrasonically extracted with Milli–Q water

(MQW, 18.2 MΩ cm). WSOM in the extracting solution was further extracted using the
typical solid phase extraction (SPE) method, as indicated by previous studies (Dittmar
et al., 2008; Qiao et al., 2020; Xie et al., 2020). Briefly, the cartridge (PPL, 0.5 g, Agilent)
was rinsed with 18 mL of methanol (LC-MS grade, Thermo Fisher) and 18 mL of HCl
solution (pH = 2) in turn. Subsequently, the extracts with acidity adjusting (pH = 2)
were injected into cartridge. Acidified MQW (18 mL) and normal MQW (6 mL) were
added in turn to remove the salt and chloride ion trapped in the cartridge. After dryness
under a stream of $N_2$, trapped OM was eluted using 15 mL of methanol. The eluted
solution was concentrated to 4 mL and then preserved at −28 °C until analysis. The
molecular compositions of WSOM in $PM_{2.5}$ samples were determined using a Bruker
Apex Ultra Fourier transform ion cyclotron resonance mass spectrometry (FT-ICR MS)
(Bruker, Germany) coupled to an Apollo II Electrospray ionization (ESI) (He et al.,





2020). The samples were injected into the ionization sources at a speed of 250 μL h$^{-1}$.
The instrument was operated in the negative-ion mode, with a spray shield voltage of
4.0 kV. The mass range was set to $m/z$ 200–800. One hundred and twenty-eight
continuous scans were acquired on each analysis to heighten the signal-to-noise ratio
of the mass spectrum. Detailed methodologies and data quality control during the
analysis were illustrated elsewhere (He et al., 2019; He et al., 2020).

Another filter cut was ultrasonically extracted with Milli–Q water for the

determination of water-soluble organic carbon (WSOC), water soluble total nitrogen
(WSTN), and inorganic ions. The mass concentrations of WSOC and WSTN in samples
were measured with a total organic carbon/total nitrogen analyser (Elementar vario,
Germany) (Xu et al., 2019). The mass concentration of WSOC was estimated to that of
WSOM using a conversion factor of 1.8 (Finessi et al., 2012; Müller et al., 2017; Simon
et al., 2011; Yttri et al., 2007). The mass concentrations of water-soluble inorganic ions,
such as $SO_4^{2-}$, $NO_3^-$, $NH_4^+$, and $K^+$, were measured using an ion chromatography
system (Dionex, Thermo, USA) (Xu et al., 2022; Xu et al., 2020b). The mass
concentration of water-soluble organic nitrogen (WSON) was calculated as the
difference in the concentrations between WSTN and inorganic nitrogen (i.e., $NO_3^-$_N
+ $NO_2^-$_N + $NH_4^+$_N) (Xu et al., 2020a). Ambient temperature and relative humidity
were measured using a temperature and humidity monitor (CEM 9880M, China).

**2.3 Data Processing and Statistical Analyses.**

The molecular formulas assigned from FT-ICR MS were classified into four main





compound groups in this study. These identified groups include CHO (containing only
C, H, and O), CHON (containing C, H, O, and N), CHOS (containing C, H, O, and S),
and CHONS (containing C, H, N, O, and S) (Song et al., 2018). The double-bond
equivalent (DBE) was calculated to describe the unsaturation degree of organic
compounds (**Supplementary Information**) (Qiao et al., 2020; Schmidt et al., 2017).
The modified aromaticity index ($AI_{mod}$) was also calculated to reflect the aromaticity of
organic molecules (**Supplementary Information**) (Koch and Dittmar, 2006; Schmidt
et al., 2017). The carbon oxidation state (OSc) can be used to indicate the evolving
composition of aerosol organics that underwent oxidation processes (Kroll et al., 2011).
The calculation of OSc was detailed in the **Supplementary Information.** According to
the oxygen-to-carbon (O/C) and hydrogen-to-carbon (H/C) elemental ratios, the
identified molecular formulas were further classified into five compound categories,
including unsaturated aliphatic-like, highly unsaturated-like, highly aromatic-like,
polycyclic aromatic-like, and saturated-like molecules (Sihui et al., 2021). These
classified compound categories were visualized in the van Krevelen diagram
(**Supplementary Information**).
The thermodynamic model ISORROPIA-II was applied to calculate the mass
concentrations of ALW driven by inorganic components (Guo et al., 2015; Tan et al.,
2017; Xu et al., 2022). The model predicts the inorganic ALW based on particle mass
concentrations of inorganic species, RH, and ambient T (**Supplementary Information**).
Particle hygroscopicity is also influenced by organics in aerosol particles (Cruz and
Pandis, 2000; Sareen et al., 2013). The impact of organic faction on aerosol water is



complex and depends on the composition of organic matter (Nguyen et al., 2016). In
this study, the mass concentration of water associated with aerosol organic fraction was
predicted    according    to    a    previously    reported    model    with    the
Zdanovskii−Stokes−Robinson mixing rule (**Supplementary Information**) (Nguyen et
al., 2015; Nguyen et al., 2016).

**3 Results and Discussion**
**3.1 Chemical Characteristics of PM$_{2.5}$ in Different Road Segments.**

**Figure 1** compares the differences in the chemical composition of PM$_{2.5}$ collected

in the air spray road segment and ground aspersion road segment. WSOM was the
dominant component irrespective of the weather and road section where the samples
were collected, which accounted for 30–40% of the water-soluble aerosols (**Figure1a–**
**c**). From March 23 to March 25, the mass concentrations and fractions of WSOM
(/WSOC) were higher in the air spray road segment than in the ground aspersion road
segment. For the case without water spray treatment (as reference group), the chemical
composition characteristics of PM$_{2.5}$ samples collected from those two adjacent road
segments only showed small differences (**Figure1d,h,i**). Obviously, these variations in
the mass concentrations and fractions of WSOM can be attributed to the difference in
water-soluble SOA yield or formation pathway caused by different water spray
treatments.

The variation pattern of ALW was similar to that of WSOC during the study

periods (**Figure1e–h**). Moreover, the mass concentrations of WSON also tended to


decrease from the air spray road segment to the ground aspersion road segment. Linear
regression analysis for all data showed that the mass concentrations of ALW were
significantly positively ($P < 0.01$) correlated with those of WSOC and WSON, with $R^2$
of 0.84 and 0.75, respectively. The results were consistent with those obtained by the
previous studies conducted in an agriculture area in Italy (Hodas et al., 2014) and a
suburban forest site in Tokyo (Xu et al., 2020a). Those studies suggested that the ALW
dependence of reactive gas uptake and subsequent aqueous reactions significantly
contributed the production of WSOC and WSON. Thus, the increase in ALW
concentration after air spraying can promote the formation of aqueous SOA in the road
environment.

Nitrate and sulfate were the most abundant inorganic components (**Figure1a–d**),

which have been identified as typical factors controlling ALW (Hodas et al., 2014).
From the air spray road segment to the ground aspersion road segment, the decrease in
nitrate concentration was more significant than that in sulfate concentration (**Figure1e–**
**f**). Moreover, the concentration of nitrate significantly correlated with that of ALW ($P$
$< 0.01$, $R^2 = 0.7$). In contrast, the sulfate did not show a strong correlation with ALW
($R^2 = 0.3$). In the region with large $NO_x$ and ammonia emissions (traffic-related) (Yang
et al., 2022), the formation of nitrate could be promoted by enhanced RH (24–43% of
increase) caused by air spraying. This is partly consistent with the thermodynamics of
ammonium nitrate formation (Hodas et al., 2014; Mozurkewich, 1993). Thus, the
increase in ALW concentration after air spraying was mainly driven by RH and locally
(traffic emissions) formed nitrate aerosol. It also implied that the formation of nitrate





and ALW is mutually reinforcing (Chen et al., 2022).
Interestingly, $Ca^{2+}$ and $Mg^{2+}$ showed a significant increase in the concentration
from the air spray road segment to the ground aspersion road segment (**Figure1i–k**),
which was contrary to the case of other components (e.g., WSOC, ALW and nitrate). In
addition, during March 26 without water spray treatment, the differences in both $Ca^{2+}$
and $Mg^{2+}$ concentrations between those two adjacent road segments were almost
negligible (**Figure1f**). It is well known that $Ca^{2+}$ and $Mg^{2+}$ are typical crustal materials
and are mainly enriched in atmospheric coarse particles (Chen and Chen, 2008). Thus,
a decrease in $Ca^{2+}$ and $Mg^{2+}$ concentrations after air spraying implied that the water mist
sprayed by mist cannon truck had a better effect on suppressing road dust than the
ground aspersion by traditional sprinkling truck.

**3.2 General Molecular Characteristics of Water-Soluble Organic Aerosols.**
Thousands of molecular formulas (5966–8102) were observed in WSOM in $PM_{2.5}$
collected from road environment (**Table 1**). The CHO molecular formulas (1089–2037)
accounted for 20–25% in all molecular formulas. CHO compounds were classified by
the number of oxygen atoms in their molecules, according to which the subgroups
ranged from $O_2$ to $O_{15}$ (**Figure 2**). The number and intensity of dominated $O_5$–$O_{10}$
subgroups accounted for 72–85% and 71–86% of the total compounds, respectively;
moreover, these percentages were higher than the results reported for aerosols in Beijing
(Xie et al., 2020). The average H/C and O/C ratios of CHO compounds varied from
1.08 to 1.24 and from 0.42 to 0.49, respectively (**Table S1**). The average O/C ratios





were higher than the value ($0.03 \pm 0.11$) obtained from typical unban aerosols (Beijing,
China), while the H/C ratios showed relatively small differences between our results
and observation results in Beijing ($1.14 \pm 0.37$) (Xie et al., 2020). These dissimilarities
might be attributed to the more complex sources of urban aerosols compared to aerosols
collected from road environment.
The average H/C and O/C ratios of CHON compounds ranged from 1.05 to 1.21
and 0.42 to 0.51, respectively (**Table S1**). The H/C ratio ranges of CHON compounds
in this study overlapped with those measured in previous studies (Sihui et al., 2021; Xie
et al., 2020). However, the O/C ratios of CHON compounds were relatively higher in
road-derived aerosols than in aerosols or snow collected in urban areas (building roof)
(Sihui et al., 2021; Xie et al., 2020), which implied the importance of source strength
(e.g., vehicle emissions) to aerosol chemical composition. The number of CHON
formulas (1501–2685) was much higher than that of CHO formulas (**Table 1**). The
assigned CHON formulas were further divided into $CHON_1$ ($N_1O_2$–$N_1O_{16}$), $CHON_2$
($N_2O_2$–$N_2O_{13}$), and $CHON_3$ ($N_3O_2$–$N_3O_{13}$) groups (**Figure 2**). $CHON_1$ was found to be
the dominant nitrogen-containing species in all samples, which was consistent with
previous reports on urban aerosols and snow (Sihui et al., 2021; Xie et al., 2020).
Moreover, the $CHON_1$ compounds with O/N > 2 contributed 99.2% – 100.0% to total
$CHON_1$ species in all samples. The $CHON_2$ compounds with O/N > 2 accounted for
90.2–100.0% of CHON species containing two nitrogen atoms. For $CHON_3$ group, the
proportion of nitrogen-containing compounds with O/N > 2 was 53.0–61.8%. The
CHON species with O/N > 2, which allows an assignment of oxidized-form nitrogen,



are preferentially ionized in negative electrospray ionization mode (Lin et al., 2015;
Sihui et al., 2021). Studies on the compositions of organic matter in urban rainwater
and aerosols have suggested that numerous CHON compounds contained oxidized
nitrogen function groups (e.g., $-ONO_2$) and that $NO_x$-related oxidation processes are
responsible for the formation of these CHON compounds (Altieri et al., 2009; Lee et
al., 2016). Thus, the CHON compounds with O/N > 2 in our $PM_{2.5}$ samples can be
assumed to be substantially in an oxidized form (e.g., organic nitrates).

**Figure 2** also shows the differences in the number of CHO and CHON species

between the air spray road segment and ground aspersion road segment. The abundance
of each $O_n$ subgroup in CHO compounds considerably enhanced after air spraying,
especially the subgroups of $O_5$–$O_{11}$. In contrast, the number of CHO species for these
two cases without water spray treatment showed a considerably small difference
(**Figure 2d**). In general, the total number of CHO compounds increased significantly
after air spraying (**Table 1** and **Figure 2**). However, there was no significant change for
the total number of CHO species between the two cases without water spray treatment.
These findings implied that increased ALW via the air spraying can contribute
substantially to the formation of CHO compounds with a more oxygenated state.

The number of CHON compounds decreased significantly from the air spray road

segment to the ground aspersion road segment, a variation pattern of which was similar
to that of CHO compounds (**Figure 2** and **Table 1**). Furthermore, the decrease in
number from the air spray road segment to the ground aspersion road segment was more
remarkable for $CHON_1$ compounds than for $CHON_2$ compounds. In contrast, the





variation in the number of $CHON_3$ after air spraying was less significant than that of
$CHON_{1-2}$ compounds. In addition, insignificant change in the number of CHON
compounds was found in $PM_{2.5}$ collected in the two road segments without water spray
(**Figure 2d**). As mentioned above, the potentially high abundance of organic nitrates
has been suggested in these road-derived aerosols. Thus, an increase in nitrogen-
containing compounds after air spraying indicated that the interactions among ALW,
traffic-derived reactive nitrogen and ambient VOCs play an important role in organic
nitrogen compound formation in aerosol fine particles. The current result can be partly
supported by that obtained by Xu et al. (2020a) in a suburban forest in Tokyo, Japan.
The authors suggested that ALW is a key driver for the formation of aerosol WSON
through secondary processes associated with atmospheric reactive nitrogen and
biogenic VOCs. For the sulfur-containing compounds, their molecular number just
showed a relatively small change after air spraying (**Figure S2**). It suggested that the
impact of ALW on sulfur-containing compound formation was weaker than that of ALW
on the formation of CHO and CHON compounds in this road environment.

**3.3 Newly Emerging CHO and CHON Species under the Influence of High ALW.**

The molecular compositions of CHO compounds in $PM_{2.5}$ in the van Krevelen

diagram were scattered across wider ranges in the air spray road segment than in the
ground aspersion road segment, particularly in the sunny days (24 and 25 March)
(**Figure 3**). The result may be attributed to the increasing molecular diversity caused by
ALW-related atmospheric processes. It also implied the importance of photochemical





reactions in CHO compound formation. Further, the unique CHO compounds were
identified between $PM_{2.5}$ samples in the air spray road segment (/no water spray road
segment (A)) and the ground aspersion road segment (/no water spray road segment (B))
(**Figure S3**). On 23 March and 24 March, the newly emerging CHO compounds after
air spraying were dominated by unsaturated aliphatic-like and highly unsaturated-like
compounds. However, both unsaturated-like species (unsaturated aliphatic-like and
highly unsaturated-like) and aromatic-like species (highly aromatic-like and polycyclic
aromatic-like) contributed significantly to the newly emerging CHO compounds after
air spraying on 25 March when the ALW and traffic flow were higher than other days
(**Figure S3**). Obviously, the formation of those unique CHO compounds was tightly
associated with increased ALW.
**Figure 4** shows the OSc of identified unique CHO molecules. The OSc values of
these CHO molecules were higher than those of primary vehicle exhausts (–2.0 to –1.9)
(Aiken et al., 2008). The OSc values of the secondary organic aerosol formed via the
reactions of anthropogenic and biogenic VOCs (e.g., isoprene, monoterpene, toluene,
alkane, and alkene) and oxidants (e.g., $O_3$ and/or •OH) varied from –1.1 to +1.0 (Kroll
et al., 2011; Li et al., 2021), which was within the OSc value ranges of CHO molecules
measured in this study. In addition, hydrocarbon-like organic aerosol (HOA) likely
linked with primary fresh vehicle exhausts (Sun et al., 2016) only accounted for less
than 6% of total unique CHO compounds (**Figure 4**). As mentioned previously, there
are dense trees on both sides of the road. Thus, these newly emerging CHO compounds
can be largely attributed to secondary processes associated with oxidation of vehicle





exhausts and biogenic VOCs by $O_3$ and/or •OH. In addition, we also observed an
oligomerization trend (e.g., methylglyoxal ($C_3H_4O_2$) to form oligomers ($C_{4-7}H_{6-10}O_5$)
in particle phase) for CHO compounds after air spraying, particularly on 25 March with
a high ALW and a large traffic flow (**Figure S4c**). The overall results implied that the
water mist sprayed by mist cannon trucks can indeed enhance the abundance and
diversity of CHO compounds in fine aerosol particles via promoting gas-to-particle
partitioning of gas-phase oxidation products of VOCs and subsequent aqueous-phase
reactions.
For the CHON compounds, their molecular compositions were scattered across an
increased ranges in the van Krevelen diagram after air spraying (**Figure S5**). Moreover,
these newly emerging CHON molecules after air spraying showed a high diversity, as
shown in **Figure S6**. The group of $CHON_1$ was the dominant nitrogen-containing
compound in the identified unique CHON compounds. On 23 March and 24 March, the
mainly unique CHON compounds emerged during air spraying were unsaturated
aliphatic-like and highly unsaturated-like nitrogen-containing species. The number of
highly aromatic-like and polycyclic aromatic-like compounds that newly emerged also
increased significantly following increased traffic flow and ALW (25 March). The
results suggested that increases in ALW after air spraying can facilitate the formation
of particle-phase nitrogen-containing compounds (Hallquist et al., 2009; Xu et al.,
2020a).
Organic nitrates have been supposed to be abundant in our $PM_{2.5}$ samples collected
from road environment. It is well documented that atmospheric organic nitrates are





primary, secondary, and tertiary byproducts of anthropogenic and biogenic VOCs with
$O_3$ (/•OH) under the influence of $NO_x$ or nitrate radicals (Lee et al., 2016; Sihui et al.,
2021; Yeh and Ziemann, 2014). In addition, numerous organic nitrates are known as
semivolatile compounds, with a partition between the gas and particle phases when they
are oxidized or photolyzed (Bean and Hildebrandt Ruiz, 2016). Recently, an oxidation
and hydrolysis mechanism associated with the atmospheric organic nitrates formation
has been proposed to interpret the potential origins or precursors of CHON compounds
(Sihui et al., 2021). In this study, we found that 68–82% of newly emerging $CHON_1$
compounds after air spraying can be explained by oxidation (e.g., $R_1OH$)-product (e.g.,
$R_1ONO_2$) pair (**Figure 5a-c**). It indicated that these newly emerging $CHON_1$
compounds were largely derived from the oxidation of CHO species under existence of
$NO_x$. A similar pattern was also observed in the newly emerging $CHON_2$ compounds
(**Figure S7**). The number of unique $CHON_1$ and $CHON_2$ compounds in the two cases
without water spray treatment was much less than that in the cases with air spraying
and ground aspersion (**Figure 5** and **Figure S7**). However, a significant oligomerization
trend for CHON compounds after air spraying was not observed after air spraying
(**Figure S8**). It should be pointed out that vehicle exhausts and roadside vegetation are
important sources for VOCs and $NO_x$ in this road environment. These results suggested
that increased ALW caused by air spraying can serve as an abundant medium for the
formation of organic nitrates via CHO compounds as potential precursors.

**3.4 Environmental Implication.**





Misting cannon sprayers are commonly applied in greening maintenance and
conventional agriculture for the distribution of fertilizers, pesticides, and herbicides. In
recent years, misting cannon trucks are developed and serve as an excellent option for
road dust control due to the production of tiny water droplets that can drop dust to the
ground. In particular, it is a high-performance system of spraying disinfection in the
road environment during the COVID-19 epidemic period. For the first time, we provide
a detailed characterization of chemical compositions in road-derived $PM_{2.5}$ under the
influence of air spraying. Recent study conducted in a rural site (China) has suggested
that gaseous water-soluble organic compounds mainly partitioned to the organic phase
under the condition of RH less than 80% (relatively low ALW) but to ALW under the
humid condition (RH > 80%, as air spraying operation), highlighting the importance of
high ALW in SOA formation processes (Lv et al., 2022). Our results verified the
formation of numerous new CHO and CHON compounds by ALW-related promoting
effect (**Figure 6**). In particular, the mass concentrations of WSOC in $PM_{2.5}$ increased
by 62–70% after air spraying. Clearly, although the air spraying by mist cannon system
could exert a better effect on suppressing road dust than the ground aspersion, as
discussed previously, air pollution induced by increased SOA will be exacerbated in the
road environment.
To further reveal the influence of air spraying on $PM_{2.5}$ pollution on the roadside,
we investigated the time series of percentage variation in $PM_{2.5}$ mass concentrations
after mist cannon truck operation at a low-speed (< 30 km/h) (**Figure 7**). At 25–35
minutes    after    the    mist    cannon    truck    passed,    the    increase    proportion    of    $PM_{2.5}$



concentration on the roadside gradually reached the maximum (~13%, on average).
Subsequently, the proportion of increase in $PM_{2.5}$ concentration gradually decreased,
reaching ~6% at 50 min after the mist cannon truck was operated. It should be noted
that the width of the road segment (81 road, Nanchang, China) where $PM_{2.5}$ was
monitored is very large (~43 m). Thus, on conventional urban roads, the water mist
sprayed by mist cannon truck should exert a greater promoting effect on the formation
of $PM_{2.5}$. The overall results suggested that mist cannon truck cannot effectively reduce
the $PM_{2.5}$ level in the road environment, but lead to aggravation of $PM_{2.5}$ pollution.
The chemical composition of fine aerosol particles in the urban road atmosphere
is highly complex, including a lot of harmful organic compounds (polycyclic aromatic
hydrocarbons and nitro-aromatics), as indicated by our measurements and previous
study (Tong et al., 2016). Emissions from vehicles and roadside greening vegetation are
important anthropogenic and biogenic sources of reactive gas-phase OC and key
precursors to form SOA in urban areas (Gentner et al., 2012; Tong et al., 2016; Xu et
al., 2020a). In particular, organic aerosol composition in the road environment can be
strongly impacted by vehicle emissions (e.g., VOCs and $NO_x$) (Tong et al., 2016).
Inhalation of the particles containing harmful organics can be responsible for a number
of adverse health effects (Künzli et al., 2000). However, the wide application of mist
cannon truck by local environmental protection department undoubtedly accelerates the
formation processes of SOA and $PM_{2.5}$ associated with ALW, which will further worsen
the urban road environment and cause health hazards to walking residents. Thus, the
present study provides the crucial information for the decision makers to regulate the





current mist cannon truck operation in many cities in China.

**4 Conclusions**
We investigated the changes in chemical compositions of $PM_{2.5}$ collected from the
road sides of the urban road (Nanchang, eastern China) with the simulated operations
of mist cannon truck and traditional sprinkling truck. Moreover, we also conducted
online measurement of $PM_{2.5}$ concentration in the urban road segment where the mist
cannon truck passed by. The mass concentrations of WSOC in $PM_{2.5}$ increased
significantly (62-70%) from the ground aspersion road segment to the air spray road
segment. Similarly, ALW, mainly driven by RH and locally formed nitrate aerosol, also
showed a significant increase after air spraying. Moreover, we found that the mass
concentration of ALW in $PM_{2.5}$ was significantly ($P < 0.01$) correlated with that of
WSOC and WSON. Thus, the increase of ALW after air spraying can promote the
formation of particle-phase water-soluble organics in the road environment. In addition,
a decrease in $Ca^{2+}$ and $Mg^{2+}$ concentrations after air spraying suggested that the water
mist sprayed by mist cannon truck exerted a better effect on suppressing road dust than
the ground aspersion by traditional sprinkling truck.
A comparison in the number of CHO and CHON species between the air spray
road segment and ground aspersion road segment suggested that the increase of ALW
after air spraying can enhance the abundance and diversity of CHO and CHON
compounds in fine aerosol particles. The newly emerging CHO compounds after air
spraying can be largely attributed to secondary processes associated with oxidation of





vehicle exhausts and biogenic VOCs by oxidants and oligomerization. Organic nitrates
were considered to be the abundant nitrogen-containing compounds in the $PM_{2.5}$
samples. Furthermore, we found that the newly emerging organic nitrates were largely
derived from the oxidation of CHO species under existence of $NO_x$.

At 25–35 minutes after the mist cannon truck passed, $PM_{2.5}$ concentration on the

roadside increased by up to 13%, on average. The proportion of increase in $PM_{2.5}$
concentration gradually decreased to ~6% at 50 min after the mist cannon truck was
operated. The overall results suggested that although mist cannon truck could exert a
better effect on suppressing road dust than the traditional sprinkling truck, air pollution
induced by increased SOA and $PM_{2.5}$ level will be exacerbated in the urban road
environment. Our findings provide new insights into the formation processes of SOA
associated with the water mist sprayed by mist cannon truck in the road atmospheric
environment.

**Data Availability**
The    data    presented    in    this    paper    can    be    accessed    at    zenodo
(DOI:10.5281/zenodo.7113675).

**Supporting Information**
Additional 1 table and 8 figures and details about parameter calculation, compound
categorization, and ALW prediction.



**Author contributions**
Conceptualization: Yu Xu, Hua-Yun Xiao, Dai-She Wu
Methodology: Yu Xu, Xin-Ni Dong, Chen He
Investigation: Xin-Ni Dong, Hong-Wei Xiao
Writing—original draft: Yu Xu
Writing—review & editing: Yu Xu

**Competing interests**
The authors declare no competing financial interest.

**Acknowledgements**
This study was kindly supported by Shanghai "Science and Technology Innovation
Action Plan" Shanghai Sailing Program through grant 22YF1418700 (Y. Xu) and the
Natural (Youth) Science Foundation of Jiangxi, China through grant 20212BAB213039
(Y. Xu).

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



**Table 1.** The number of compounds in different subgroups in different samples and the number fractions of common molecules in the same subgroup in different samples.

| Sample type (Date: Mar. 23–26) | Total | CHO | Common (CHO) | Common fraction (CHO) | CHON | Common (CHON) | Common fraction (CHON) |
|---|---|---|---|---|---|---|---|
| Air spray (23) | 7069 | 1766 | 1104 | 63% | 2375 | 1308 | 55% |
| Ground aspersion (23) | 6166 | 1233 | | 90% | 1803 | | 73% |
| Air spray (24) | 7317 | 1861 | 990 | 53% | 2447 | 1225 | 50% |
| Ground aspersion (24) | 6067 | 1098 | | 90% | 1501 | | 82% |
| Air spray (25) | 8102 | 2037 | 804 | 39% | 2685 | 1209 | 45% |
| Ground aspersion (25) | 5966 | 845 | | 95% | 1464 | | 83% |
| No water spray (A) (26) | 6998 | 1621 | 1315 | 81% | 2120 | 1579 | 74% |
| No water spray (B) (26) | 6990 | 1539 | | 85% | 1963 | | 80% |





**Figures 1–7**
**Figure 1.**

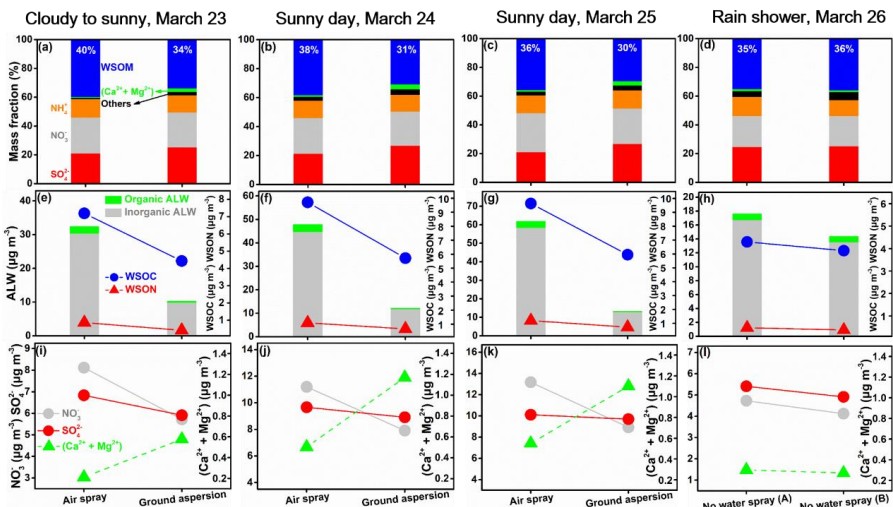

**Figure 1.** The mass fractions of the chemical components in PM2.5: (a, b, and c) air
spray road segment vs ground aspersion road segment and (d) no water spray road
segment (A) vs no water spray road segment (B). The mass concentrations of nitrate,
ammonium, and the sum of calcium and magnesium in PM2.5: (e, f, and g) air spray
road segment vs ground aspersion road segment and (h) no water spray road segment
(A) vs no water spray road segment (B). The mass concentrations of ALW, WSOC, and
WSOM in PM2.5: (i, j, and k) air spray road segment vs ground aspersion road segment
and (l) no water spray road segment (A) vs no water spray road segment (B).





**Figure 2.**

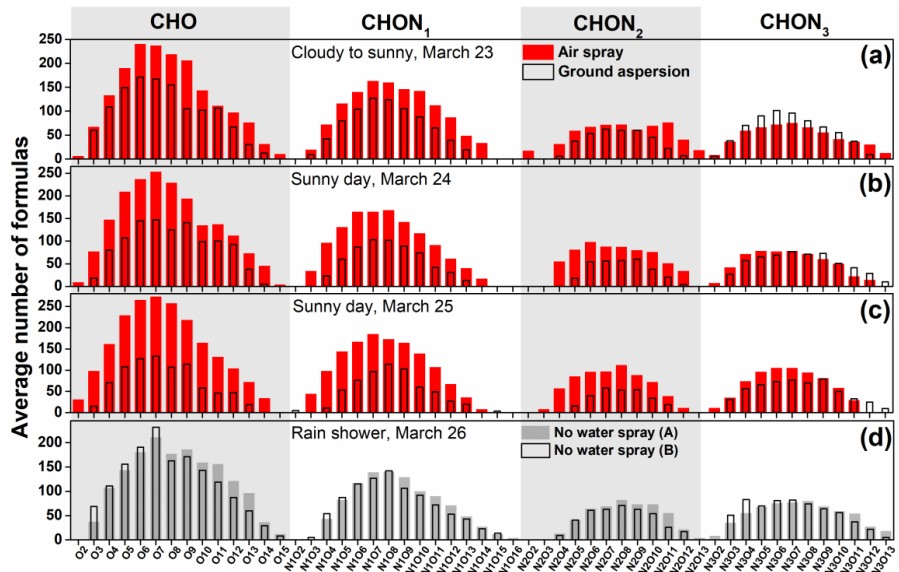

**Figure 2.** Classification of CHO and CHON species into subgroups according to the
number of O atoms in their molecules in WSOM in PM$_{2.5}$ collected from different cases:
(a, b, and c) air spray road segment vs ground aspersion road segment and (d) no water
spray road segment (A) vs no water spray road segment (A).



**Figure 3**

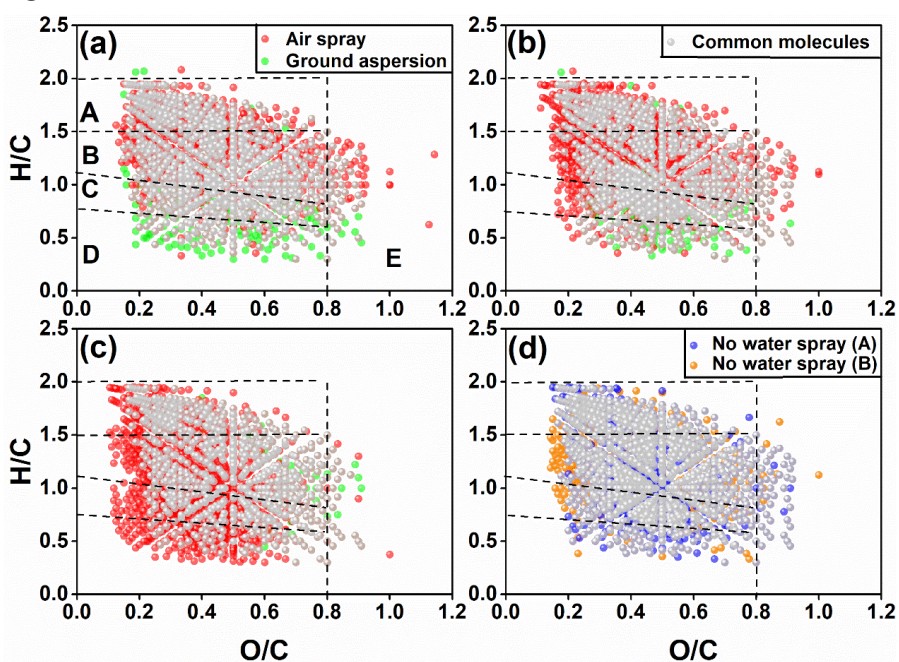

**Figure 3.** Van Krevelen diagrams of CHO compounds in WSOM in PM$_{2.5}$ collected
from different cases: air spray road segment vs ground aspersion road segment on (a)
March 23, (b) March 24, and (c) March 25 and two road segments without water spray
(A vs B) on (d) March 26. The circles of different colors indicate the unique organic
compounds identified in the above cases of paired comparison. Common molecules
identified in different cases are shown as gray circles. The classifications of compounds
include (A) unsaturated aliphatic-like, (B) highly unsaturated-like, (C) highly aromatic-
like, (D) polycyclic aromatic-like, and (E) saturated-like molecules.



**Figure 4.**

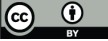

**Figure 4.** $OS_c$ of unique CHO molecules in WSOM in $PM_{2.5}$ collected from different
cases: air spray road segment vs ground aspersion road segment on (a) March 23, (b)
March 24, and (c) March 25 and two road segments without water spray (A vs B) on (d)
March 26. For the above cases of paired comparison, the unique CHO compounds
indicate the CHO molecules identified in $PM_{2.5}$ collected from the air spray (/no water
spray-A) road segments. The light orange background represents areas of HOA
(hydrocarbon-like organic aerosol), BBOA and VEOA (biomass burning and vehicle
emission organic aerosols) (Kroll et al., 2011; Tong et al., 2016), SV-OOA (semivolatile
oxidized organic aerosol), and LV-OOA (low-volatility oxidized organic aerosol) (Kroll
et al., 2011).

**Figure 5.**




**Figure 5.** $OS_c$ of unique $CHON_1$ molecules in WSOM in $PM_{2.5}$ collected from different
cases: air spray road segment vs ground aspersion road segment on (a) March 23, (b)
March 24, and (c) March 25 and two road segments without water spray (A vs B) on (d)
March 26. For the above cases of paired comparison, the unique $CHON_1$ compounds
indicate the $CHON_1$ molecules identified in $PM_{2.5}$ collected from the air spray (/no
water spray-A) road segments. The light orange background represents areas of HOA,
BBOA and VEOA, SV-OOA, and LV-OOA. The grey circles refer to the identified
oxidation-product pairs.








**Figure 6.**

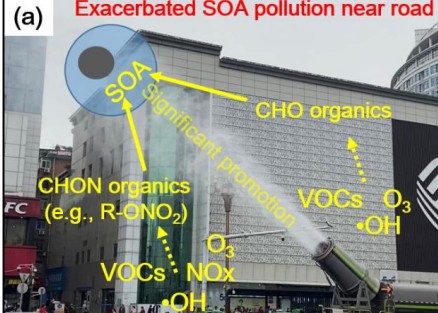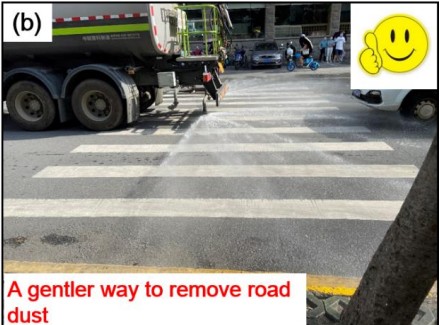


**Figure 6.** Conceptual picture showing the influence of (a) mist cannon truck and (b)
traditional sprinkling truck on SOA formation in the urban road environment.






**Figure 7.**

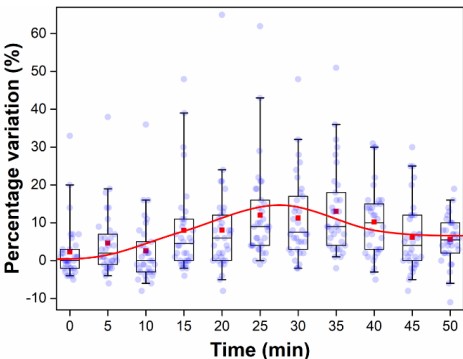


**Figure 7.** The time series of percentage variations in $PM_{2.5}$ mass concentrations after
mist cannon truck operation. Each box encompasses the 25th−75th percentiles.
Whiskers are the 5th and 95th percentiles. The solid lines and squares inside boxes
indicate the median and mean. All individual data are also presented as circles.