# Peer review of "Formation of Water-Soluble Organic Aerosol and"

_Atmospheric Chemistry and Physics, 2022_

## Author Comment (AC1)

**General.**

We would like to appreciate the editor and reviewers for providing the valuable comments and a better perspective on our work to improve the manuscript. In particular, we are very grateful to the editor and reviewers for giving us the opportunity to make revision. We have revised our manuscript by fully taking the editor's and reviewers' comments into account. Responses to specific comments raised by the editor and reviewers are described below. All the changes made and appeared in the revised text are shown in red. All detailed answers to comments are displayed in blue.

**Comments of Reviewer #2 and our responses to them**

Comments:

*This study investigated the changes in aerosols caused by mist cannon trucks. The topic is interesting and practical in China and fits the scope of the ACP journal. However, some concerns need to be addressed before this manuscript can be considered for publication.*

Response: We appreciate the reviewer's valuable comments on our work. Our responses to the specific comments and changes made in the manuscript are given below.

Major Comments:

*1) The manuscript needs significant improvement in English editing and figures (including captions). A native English speaker should review the manuscript. For example, in Figure 3, the points in the plot are overlapping, making it difficult to distinguish. The "Air spray" and "ground aspersion road" segments were marked as "(A)" and "(B)," which is the same*

*as the compound classification "(A) unsaturated aliphatic and (B) unsaturated-like." "Common molecules" were marked only in the (b) panel but also used in a, c, and d. Such issues must be fully addressed before the manuscript can be considered for further processing.*

Response: We appreciate the reviewer's kind and valuable comments. In particular, we are very sorry for the confusion caused by our writing.

---For the comment: A native English speaker should review the manuscript.

Answer: The manuscript has been reviewed by a native English speaker (we have expressed our gratitude to the person who helped us edit the language in the acknowledgement section).

---For the comment: In Figure 3, the points in the plot are overlapping, making it difficult to distinguish.

Answer: The overlapped points in Figure 3 (now as Figure 4 in the revised manuscript) indicate common CHO molecules in WSOM in $PM_{2.5}$ collected from different cases. We want to show that the molecular compositions of CHO compounds in $PM_{2.5}$ in the van Krevelen diagram were scattered across wider ranges in the air spray road segment than in the ground aspersion road segment, particularly in the sunny days (24 and 25 March)

More descriptions to support the above results were shown below (Lines 351–355).

Lines 351–355: …Moreover, common CHO molecules accounted for 39% (sunny day) – 63% (cloudy to sunny day) and 90–95% of CHO molecules in $PM_{2.5}$ collected from the air spray and ground aspersion road segments, respectively (Table 1). In contrast,

common CHO molecules contributed 81–85% of CHO molecules in $PM_{2.5}$ collected from two road segments without water spray (I vs II)…

---For the comment: The "Air spray" and "ground aspersion road" segments were marked as "(A)" and "(B)," which is the same as the compound classification "(A) unsaturated aliphatic and (B) unsaturated-like." "Common molecules" were marked only in the (b) panel but also used in a, c, and d.

Answer: All confusing marks or symbols have been revised. For example: two road segments without water spray were marked as "I" and "II", respectively; and the classifications of compounds include unsaturated aliphatic-like (UA), highly unsaturated-like (HU), highly aromatic-like (HA), polycyclic aromatic-like (PA), and saturated-like (Sa) molecules.

*2) The authors assumed that the background $PM_{2.5}$ concentrations in the two sampling sites are the same. However, this assumption needs to be validated by comparing the two samplers and the meteorology. Since high-volume samplers can easily be influenced by changes in flow rate and street canyon effects can cause large differences between the two sampling sites, the authors should provide more information and discussion on those aspects. The dispersion effects and changes in traffic flow need to be considered when making the statement that the $PM_{2.5}$ increase 20 minutes later is due to the mist trucks.*

Response: This is a crucial question. We are very grateful to the reviewers for this excellent comment. We would like to explain this comment from the following two aspects

1) Two research sites are located on the same road without forks. Moreover, we have set up a reference group, as shown in Figures 2d,h,l. When there was no spraying activity, the concentrations of WOSC, ALW, and other components as well as water-soluble organic compound compositions in $PM_{2.5}$ between these two study sites showed a relatively small difference. Thus, the impact of background $PM_{2.5}$ or meteorological factor on $PM_{2.5}$ compositions or levels was expected to be similar between these two study sites.

2) After the misting cannon truck passed through the monitoring point several times, the $PM_{2.5}$ online monitoring ($n = 34$, within a month) was performed to avoid (or decrease) the effect of resuspended road dust (tightly associated with the dispersion effect and traffic flow). Accordingly, the resuspension of road dust was expected to exert a relatively minor impact on the $PM_{2.5}$ level near road. The concentrations of $PM_{2.5}$ showed an increasing trend after the mist cannon truck passed the monitoring point for 15 minutes. Thus, the water droplets sprayed by the mist cannon truck cannot directly cause an increase in $PM_{2.5}$ concentration, suggesting that the increased $PM_{2.5}$ should be secondarily formed after water mist spraying (~15 minutes). This consideration was also supported by a significant increase in the concentration and number of water-soluble organic compounds after air spraying (Fig. 2 and Fig. 3). Thus, the overall results can indicate that our statement is reliable.

The added descriptions in the revised manuscript were shown below (Lines 225–227 and 452–464).

Lines 225–227: …It suggested that the impact of background $PM_{2.5}$ or meteorological factor on $PM_{2.5}$ composition or level was similar between these two study sites…

Lines 452–464: …It should be pointed out that misting cannon trucks usually operate back and forth on specific road sections to prevent the resuspension of dust. After the misting cannon truck passed through the monitoring site several times, repeated online PM$_{2.5}$ monitoring ($n = 34$, within a month) was performed to exclude the impact of dispersion and traffic flow on analysis results. Accordingly, the resuspension of road dust was expected to exert a relatively minor impact on the PM$_{2.5}$ level near road. The concentration of PM$_{2.5}$ showed an increasing trend after the mist cannon truck passed the monitoring point for 15 minutes. Thus, the water droplets sprayed by the mist cannon truck cannot directly cause an increase in PM$_{2.5}$ concentration, suggesting that the increased PM$_{2.5}$ should be secondarily formed after water mist spraying (~15 minutes). This consideration was also supported by a significant increase in the concentration and number of water-soluble organic compounds after air spraying (Fig. 2 and Fig. 3)…

*3) The results are based on qualitative information from FT-ICR MS, so statements such as the increase of SOA should be toned down. Also, the size distribution of the droplets between the two types of water spray can cause a large discrepancy between the two samplers with two PM$_{2.5}$ inlets, as the fog droplet caused by the mist cannon trucks can reach a size of >1 um. The authors should add more discussions on the different sizes of droplet particles.*

Response: Partitioning gaseous water-soluble organic compounds to particle phase and subsequent reactions in particle phase are the major formation pathways of atmospheric secondary organic aerosols (SOA) (Lv et al., 2022). Thus, SOA  has been replaced with

water-soluble organic compounds in the revised manuscript.

For the online monitoring of $PM_{2.5}$ mass concentration, the results suggested that the concentration of $PM_{2.5}$ showed an increasing trend after the mist cannon truck passed the monitoring point for 15 minutes. Thus, the water droplets sprayed by the mist cannon truck cannot directly cause an increase in $PM_{2.5}$ concentration. This means that the increased $PM_{2.5}$ should be secondarily formed after water spraying. This consideration was also supported by a significant increase in the concentration and number of water-soluble organic compounds after air spraying (Fig. 2 and Fig. 3). Overall, droplet size is not the main factor affecting $PM_{2.5}$ levels in this study. This may be because the strong light and high temperature during the operation of the mist cannon truck can cause rapid evaporation of water droplets.

More explanations have been added in the revised manuscript, as mentioned in comment 2 (Lines 452–464).

*4) The authors should provide more evidence for the statement that the enhanced $NO_3$ formation with high ALW. The heterogeneous reaction is typically more critical during the nighttime (lines 244-245). Additionally, the dependence of the $N_2O_5$ take-up coefficient and RH was found to be weak after RH>50% (Wang et al., 2020). In this study, RH is usually much higher (~80%).*

Response: In polluted regions, the nitrate aerosol is mainly derived from the following two pathways: (1) the gas-phase oxidation of nitrogen dioxide ($NO_2$) by hydroxyl radical ($\bullet OH$) to form nitric acid ($HNO_3$) (Calvert and Stockwell, 1983) and (2) the

heterogeneous hydrolysis of dinitrogen pentoxide ($N_2O_5$) that was produced from the gas-phase reaction of $NO_2$ with nitrate radical ($NO_3\bullet$) on aqueous aerosols (Bertram et al., 2009; Wagner et al., 2013; McDuffie et al., 2019). The gas-phase $\bullet OH + NO_2$ pathway primarily occurs during the daytime and is mainly influenced by the atmospheric oxidation capacity (Chen et al., 2020; Fu et al., 2020). In this study, the field experiments were conducted during daytime (9:00−13:00 LT) with strong sunlight. Moreover, RH is relatively high. Thus, $O_3$ can be rapidly photolyzed to form $\bullet OH$ under conditions with abundant $H_2O$ and sunlight (Li et al., 2022). In particular, we found that the concentration of nitrate significantly correlated with that of ALW ($P < 0.01$, $R^2 = 0.7$). The overall results suggested that the increase in nitrate concentration after air spraying was tightly with increased ALW.

The added descriptions in the revised manuscript were shown below (Lines 250–257).

Lines 250–257: …As we know, the gas-phase oxidation of $NO_2$ by hydroxyl radical ($\bullet OH$) to form nitric acid ($HNO_3$) is an important pathway for the formation of daytime nitrate aerosol (Fu et al., 2020; Chen et al., 2020). Hydroxyl radical can be rapidly produced by $O_3$ photolysis under conditions with abundant water vapour and sunlight (as in this study) (Li et al., 2022), which is undoubtedly beneficial for the production of $HNO_3$. Thus, in the region with large $NO_x$ and ammonia emissions (originated from vehicle exhausts (Yang et al., 2022)), the formation of daytime nitrate aerosol could be promoted by enhanced RH (24–43% of increase) caused by air spraying…

*5) The O/C ratios (0.03) in Beijing listed are far lower than the current understanding, which is around 0.3-0.7 (Hu et al., 2017). The authors need to explain this large difference.*

Response: We apologize for the writing error. The correct O/C ratio should be 0.33 ±
0.11. The revision has been made in the revised manuscript. In addition, a new case (Hu
et al., 2017) has been added in the revised manuscript.

Line 284: … 0.33 ± 0.11…

Lines 286–289: … In addition, another study performed in Beijing showed that the
average O/C and H/C ratios of organic aerosols were in the range of 0.47–0.53 and
1.52–1.63, respectively (Hu et al., 2017) …

*6) The authors should explain why after the air spray, there were more high C number
compounds with lower OS compounds, as shown in Fig S4 a, b, and c. The authors should
also consider the difference in the k value for SOA and POA in calculations (Zhao et al.,
2015).*

Response: The study by Zhao et al. (2015) suggested that smaller particle had a higher κ
and a higher degree of oxidation. We know the κ value is very important for the
hygroscopicity of particles. However, in this study, we want to show that aqueous-phase
oligomerization is a potential path of water-soluble organic compound formation (Fig.
S4 a, b, and c; now Fig. S3 a, b, and c in the revised manuscript) (Ma et al., 2022).
Because increased ALW may promote the oligomerization of CHO compounds, higher
C number of compounds can correspond to lower carbon oxidation state (OSc).

Details were shown in the revised manuscript (Lines 382–384).

Lines 382–384: …we also observed increased oligomerization (e.g., methylglyoxal

($C_3H_4O_2$) to form oligomers ($C_{4-7}H_{6-10}O_5$) in particle phase) of CHO compounds after air spraying (Ma et al., 2022) ...

Minor Comments:

*1) Line 455: If only 4 data points were used, such a significance test is not solid anymore since it depends on normally distributed data.*

Response: We are very sorry for confusion caused by our description. In the main text, we have indicated that linear regression analysis between ALW ($n = 8$) and WSOC ($n = 8$) and WSON ($n = 8$) was conducted based on 4-day data ($R^2 = 0.75 - 0.84$, $P < 0.01$).

The number of data points has been added in the revised manuscript (Lines 234 and 497).

Line 234: …Linear regression analysis for all data showed that the mass concentrations of ALW ($n = 8$) …

Line 497: …we found that the mass concentration of ALW ($n = 8$) …

*2) Line 379 and Line 385: It should be cited as Su et al., 2021, not Sihui et al., 2021.*

Response: Thank you very much for your careful review. The revisions have been made in the revised manuscript (Lines 293, 296, 308, 408, and 413).

*3) Line 396: What does "abundant medium" mean? It should be rephrased.*

Response: We are very sorry for confusion caused by our description. We want to express that ALW is an abundant medium (Nguyen et al., 2016) that can enhance secondary organic aerosol (SOA) formation. In the revised manuscript, we have deleted the phrase and rewritten the sentence.

Lines 425–426: …the increase in ALW caused by air spraying can facilitate the formation of organic nitrates…

**Once again, we deeply appreciate the time and effort you've spent in reviewing our manuscript.**

**References**

Bertram, T. H., Thornton, J. A., Riedel, T. P., Middlebrook, A. M., Bahreini, R., Bates, T. S., Quinn, P. K., and Coffman, D. J.: Direct observations of N2O5 reactivity on ambient aerosol particles, Geophys. Res. Lett., 36, L19803, https://doi.org/10.1029/2009GL040248, 2009.

Chen, X., Wang, H., Lu, K., Li, C., Zhai, T., Tan, Z., Ma, X., Yang, X., Liu, Y., Chen, S., Dong, H., Li, X., Wu, Z., Hu, M., Zeng, L., and Zhang, Y.: Field Determination of Nitrate Formation Pathway in Winter Beijing, Environ. Sci. Technol., 54, 9243–9253, https://doi.org/10.1021/acs.est.0c00972, 2020.

Calvert, J. G. and Stockwell, W. R.: Acid generation in the troposphere by gas-phase chemistry, Environ. Sci. Technol., 17, 428–443, 1983.

Fu, X., Wang, T., Gao, J., Wang, P., Liu, Y., Wang, S., Zhao, B., and Xue, L.: Persistent Heavy Winter Nitrate Pollution Driven by Increased Photochemical Oxidants in

Northern China, Environ. Sci. Technol., 54, 3881–3889, https://doi.org/10.1021/acs.est.9b07248, 2020.

Hu, W., Hu, M., Hu, W. W., Zheng, J., Chen, C., Wu, Y. S., and Guo, S.: Seasonal variations in high time-resolved chemical compositions, sources, and evolution of atmospheric submicron aerosols in the megacity Beijing, Atmospheric Chemistry and Physics, 17, 9979-10000, 10.5194/acp-17-9979-2017, 2017.

Li, X., Zhang, Y., Shi, L., Kawamura, K., Kunwar, B., Takami, A., Arakaki, T., and Lai, S.: Aerosol Proteinaceous Matter in Coastal Okinawa, Japan: Influence of Long-Range Transport and Photochemical Degradation, Environmental Science & Technology, 56, 5256-5265, 10.1021/acs.est.1c08658, 2022.

Lv, S., Wang, F., Wu, C., Chen, Y., Liu, S., Zhang, S., Li, D., Du, W., Zhang, F., Wang, H., Huang, C., Fu, Q., Duan, Y., and Wang, G.: Gas-to-Aerosol Phase Partitioning of Atmospheric Water-Soluble Organic Compounds at a Rural Site in China: An Enhancing Effect of NH3 on SOA Formation, Environ. Sci. Technol., 56, 3915-3924, 10.1021/acs.est.1c06855, 2022.

McDuffie, E. E., Womack, C. C., Fibiger, D. L., Dube, W. P., Franchin, A., Middlebrook, A. M., Goldberger, L., Lee, B. H., Thornton, J. A., Moravek, A., Murphy, J. G., Baasandorj, M., and Brown, S. S.: On the contribution of nocturnal heterogeneous reactive nitrogen chemistry to particulate matter formation during wintertime pollution events in Northern Utah, Atmos. Chem. Phys., 19, 9287–9308, https://doi.org/10.5194/acp19-9287-2019, 2019.

Ma, W., Zheng, F., Zhang, Y., Chen, X., Zhan, J., Hua, C., Song, B., Wang, Z., Xie, J., Yan, C., Kulmala, M., and Liu, Y.: Weakened Gas-to-Particle Partitioning of Oxygenated Organic Molecules in Liquified Aerosol Particles, Environmental Science & Technology Letters, 10.1021/acs.estlett.2c00556, 2022.

Nguyen, T. K. V., Zhang, Q., Jimenez, J. L., Pike, M., and Carlton, A. G.: Liquid water: ubiquitous contributor to aerosol mass, Environ. Sci. Tech. Let., 3, 257-263. https://doi.org/210.1021/acs.estlett.1026b00167, 2016.

Wagner, N., Riedel, T., Young, C., Bahreini, R., Brock, C., Dubé, W., Kim, S., Middlebrook, A., Öztürk, F., and Roberts, J.: N2O5 uptake coefficients and nocturnal NO2 removal rates determined from ambient wintertime measurements, J. Geophys. Res.-Atmos., 118, 9331–9350, 2013.

Wang, H., Chen, X., Lu, K., Tan, Z., Ma, X., Wu, Z., Li, X., Liu, Y., Shang, D., Wu, Y., Zeng, L., Hu, M., Schmitt, S., Kiendler-Scharr, A., Wahner, A., and Zhang, Y.: Wintertime N(2)O(5) uptake coefficients over the North China Plain, Sci Bull (Beijing), 65, 765-774, 10.1016/j.scib.2020.02.006, 2020.

Yang, D., Zhu, S., Ma, Y., Zhou, L., Zheng, F., Wang, L., Jiang, J., and Zheng, J.: Emissions of Ammonia and Other Nitrogen-Containing Volatile Organic Compounds from Motor Vehicles under Low-Speed Driving Conditions, Environ. Sci. Technol., 56, 5440-5447, 10.1021/acs.est.2c00555, 2022.

Zhao, D. F., Buchholz, A., Kortner, B., Schlag, P., Rubach, F., Kiendler-Scharr, A., Tillmann, R., Wahner, A., Flores, J. M., Rudich, Y., Watne, Å. K., Hallquist, M., Wildt, J., and Mentel, T. F.: Size-dependent hygroscopicity parameter (κ) and chemical composition of secondary organic cloud condensation nuclei, Geophysical Research Letters, 42, 10,920-910,928, 10.1002/2015gl066497, 2015.

---

## Author Comment (AC3)

**General.**

We would like to appreciate the editor and reviewers for providing the valuable comments and a better perspective on our work to improve the manuscript. In particular, we are very grateful to the editor and reviewers for giving us the opportunity to make revision. We have revised our manuscript by fully taking the editor's and reviewers' comments into account. Responses to specific comments raised by the editor and reviewers are described below. All the changes made and appeared in the revised text are shown in red. All detailed answers to comments are displayed in blue.

**Comments of Referee #1 and our responses to them**

Comments:

*The paper presents a comprehensive analysis of the impacts of mist cannon use on air quality in urban areas. While the paper could benefit from improved clarity, the chemical specification seems sound, and the analysis is detailed and thorough. The results of the study are impactful, highlighting the negative effects of mist cannon use on air quality and demonstrating the potential for mist cannons to exacerbate secondary organic aerosol formation and $PM_{2.5}$ pollution in the road environment. These findings have significant implications for future research and policy aimed at improving air quality in urban areas, making this a valuable contribution to the field of air pollution research.*

Response: We appreciate the reviewer's valuable comments on our work. Our responses to the specific comments and changes made in the manuscript are given below.

Specific comments:

*1) Row 39: The statement about the 13% increase in PM$_{2.5}$ refers to the major effect on health hazard. However, the new compounds potentially formed could be more toxic. The composition change may be more important than the increase in PM$_{2.5}$. Toxicological studies could provide additional information.*

Response: We are very sorry for this inappropriate expression. In particular, we greatly appreciate your suggestion. However, due to the research on toxicology seems to deviate from the theme of this study, we have revised the relevant expression.

The relevant revision has been made in the revised manuscript (Lines 39−42).

Lines 39−42: …Thus, the application of mist cannon trucks potentially worsens the road atmospheric environment through the increase in PM$_{2.5}$ level and the production of a large number of water-soluble organic compounds in PM$_{2.5}$…

*2) Row 42: This paper lacks proper argumentation why these mist cannon trucks are used in urban environment; how much are the benefits in road dust control?*

Response: More descriptions on why the mist cannon trucks are used in urban environment have been added in the revised manuscript. However, specific reports or experimental studies about why and how mist cannon trucks can facilitate road dust control are very scarce. Thus, we conducted this research.

The relevant explanations have been added in the revised manuscript (Lines 57−58 and 63−65).

Lines 57−58: …although the relevant argumentation work is rarely systematically studied or reported…

Lines 63−65: …no study has investigated whether and how the water mist sprayed by mist cannon truck affects the road atmospheric environment…

*3) Row 56: Related to previous comment, what are the quantitative benefits of mist cannon use in emission control?*

Response: We fully understand your consideration and deeply appreciate your comment. However, as mentioned above, although the mist cannon trucks are considered to be more water-saving and efficient than the traditional sprinkling truck (the ground aspersion), the relevant argumentation work is rarely systematically studied or reported. To our knowledge, this is the first investigation about the effect of mist cannon truck on the formation of water-soluble organic compounds and the pollution control of $PM_{2.5}$.

*4) Rows 100-: The experimental description of different roads, sampling sites etc. is difficult to follow based on this text. Insertion of Figure S1 to the main text would help to clarify what has been done.*

Response: We appreciate the reviewer's kind suggestion. The Figure S1 has been added in the main text as new Figure 1.

*5) Row 109: Can you speciate here how large is the biogenic/traffic VOC roles in your dataset?*

Response: Due to the fact that the structure of the determined molecular formula cannot be identified by our analytical method, it is difficult to evaluate the relative importance of biogenic and traffic VOCs in this study. However, this will not affect the main conclusions that were drawn in this study.

*6) Row 116: I assume oral fluids are something else than water. Is this a valid background information for your analysis?*

Response: We totally agree with your point of view. Thus, the relevant description has been deleted in the revised manuscript.

*7) Row 158: Using a noun other than "speed" would be more appropriate in this case.*

Response: The sentence has been rewritten in the revised manuscript.

Lines 164–165: ... The samples were injected into the ionization source at 250 µL h$^{-1}$ through a syringe pump…

*8) Rows 402-404: It is unclear if the dust stays in the ground or if there are road cleaning*

*practices to prevent resuspension. Additional information on this would be helpful.*

Response: We thank for this important comment. Misting cannon trucks usually operate back and forth on specific road sections to prevent the resuspension of dust. Thus, we need to continuously record the variations in PM$_{2.5}$ mass concentrations near road after the mist cannon truck passed by (Figure 8). More descriptions have been added in the revised manuscript (Lines 452–456).

Lines 451–456: …It should be pointed out that misting cannon trucks usually operate back and forth on specific road sections to prevent the resuspension of dust. After the misting cannon truck passed through the monitoring site several times, repeated online PM$_{2.5}$ monitoring ($n = 34$, within a month) was performed to exclude the impact of dispersion and traffic flow on analysis results…

*9) Row 408: It is unclear what is meant by the "organic" phase in this case.*

Response: Several studies have suggested that aerosols can exist in a phase-separated form with an organic shell and an inorganic core even at an RH higher than 80% (Li et al., 2021; Ushijima et al., 2021; Yu et al., 2019). Thus, "organic phase" refers to organic shell in aerosols.

Details were shown in the revised manuscript (Lines 441–443).

Lines 441–443: …This is because aerosols can exist in a phase-separated form with an inorganic core and an organic shell (Yu et al., 2019; Li et al., 2021a; Ushijima et al., 2021) …

*10) Row 428: A fraction of road dust is likely in PM$_{2.5}$. It would be helpful to know how much new ambient PM$_{2.5}$ is produced by this operation.*

Response: Misting cannon trucks operated back and forth on investigated road sections to prevent the resuspension of dust, as mentioned above. After the misting cannon truck passed through the monitoring point several times, the PM$_{2.5}$ online monitoring was performed to avoid (or decrease) the effect of resuspended road dust. Thus, the resuspension of road dust was expected to exert a relatively minor impact on the PM$_{2.5}$ level near road. The concentration of PM$_{2.5}$ showed an increasing trend after the mist cannon truck passed the monitoring point for 15 minutes. Thus, the water droplets sprayed by the mist cannon truck cannot directly cause an increase in PM$_{2.5}$ concentration, suggesting that the increased PM$_{2.5}$ should be secondarily formed after water mist spraying. This consideration was also supported by a significant increase in the concentration and number of water-soluble organic compounds after air spraying (Fig. 2 and Fig. 3). At 25–35 minutes after the mist cannon truck passed the monitoring point, the increase proportion of PM$_{2.5}$ concentration on the roadside gradually reached the maximum (~13%, on average). Thus, these new ambient PM$_{2.5}$ is secondarily produced by mist cannon truck operation.

More explanations have been added in the revised manuscript (Lines 452–464).

Lines 452–464: …It should be pointed out that misting cannon trucks usually operate back and forth on specific road sections to prevent the resuspension of dust. After the misting cannon truck passed through the monitoring site several times, repeated online PM$_{2.5}$ monitoring ($n = 34$, within a month) was performed to exclude the impact of dispersion and traffic flow on analysis results. Accordingly, the resuspension of road

dust was expected to exert a relatively minor impact on the $PM_{2.5}$ level near road. The concentration of $PM_{2.5}$ showed an increasing trend after the mist cannon truck passed the monitoring point for 15 minutes. Thus, the water droplets sprayed by the mist cannon truck cannot directly cause an increase in $PM_{2.5}$ concentration, suggesting that the increased $PM_{2.5}$ should be secondarily formed after water mist spraying (~15 minutes). This consideration was also supported by a significant increase in the concentration and number of water-soluble organic compounds after air spraying (Fig. 2 and Fig. 3)…

*11) Figure 1: The fonts are extremely small and should be enlarged for readability.*

Response: The fonts of Figure 1 (now as Figure 2) have been increased as much as possible.

*12) Figure 1: Would there be reference data of "no air spray" & "no ground aspersion" to be added as third bars? This would be the reference to compare with in order to understand the effect of an action.*

Response: We are very grateful to the reviewers for this excellent suggestion. In this study, we have set up a reference group, as shown in Figures 2d,h,l. When there was no spraying activity, the concentrations of WOSC, ALW, and other components as well as water-soluble organic compound compositions in $PM_{2.5}$ between these two study sites showed a relatively small difference. Thus, although reference data of "no air spray" and "no ground aspersion" to be added as third bars will be very meaningful, the direct

comparison between air spray case and ground aspersion case is sufficient to achieve the purpose of this study.

*13) Figure 4: What does it mean that most points outside different categories (ellipses)?*

Response: Figure 4 (now as Figure 5 in the revised manuscript) shows OSc values of unique CHO molecules in WSOM in $PM_{2.5}$ collected from different cases. Based on existing classification method, it is not possible to classify all CHO molecules in Figure 5. However, these unique CHO molecules can at least suggest that the water mist from air spraying can indeed enhance the abundance and diversity of CHO compounds in $PM_{2.5}$.

More descriptions have been added in the revised manuscript (Lines 376–379).

Lines 376–379: …Although it is difficult to classify all CHO molecules in Fig. 5, these identified unique CHO molecules can at least suggest that the water mist from air spraying can promote the formation of CHO compounds and increase their molecular diversity…

*14) Figure 6: The format of the figure (smiley face, comments) is more suitable for a PowerPoint presentation than a formal research article.*

Response: The Figure 6 (now as Figure 7 in the revised manuscript) has been updated. Smiley face and comments have been deleted.

*15) Figure 7: It is unclear what the red continuous line represents. Additional information on this would be helpful.*

Response: The Figure 7 (now as Figure 8 in the revised manuscript) has been updated. Now, we use the red dashed line to represent the change in the average concentration of PM$_{2.5}$.

**At last, we deeply appreciate the time and effort you've spent in reviewing our manuscript.**

**References**

Li, W. J.; Teng, X. M.; Chen, X. Y.; Liu, L.; Xu, L.; Zhang, J.;Wang, Y. Y.; Zhang, Y.; Shi, Z. B. Organic Coating Reduces Hygroscopic Growth of Phase-Separated Aerosol Particles. *Environ. Sci. Technol.* 2021, 55, 16339−16346.

Ushijima, S. B.; Huynh, E.; Davis, R. D.; Tolbert, M. A. Seeded Crystal Growth of Internally Mixed Organic-Inorganic Aerosols: Impact of Organic Phase State. *J. Phys. Chem. A* 2021, 125, 8668−8679.

Yu, H.; Li, W. J.; Zhang, Y. M.; Tunved, P.; Dall'Osto, M.; Shen, X. J.; Sun, J. Y.; Zhang, X. Y.; Zhang, J. C.; Shi, Z. B. Organic coating on sulfate and soot particles during late summer in the Svalbard Archipelago. *Atmos. Chem. Phys.* 2019, 19, 10433−10446.